# Methodology and Tools to Integrate Industry 4.0 CPS into Process Design and Management: ISA-88 Use Case

**Ander Garcia [1,*], Xabier Oregui [1], Unai Arrieta [1] and Iñigo Valverde [2]**

[1]  Vicomtech Foundation, Basque Research and Technology Alliance (BRTA), Mikeletegi 57,
    20009 Donostia-San Sebastián, Spain; xoregui@vicomtech.org (X.O.); uarrieta@vicomtech.org (U.A.)
[2]  ES Solidos Process Engineering SL, Loiolako Inazio Hiribidea 27, 20730 Azpeitia, Spain; valverde@essolidos.es
*   Correspondence: agarcia@vicomtech.org

**Abstract:** Once an industrial process is designed, the real implementation of the process control is programmed into Supervisory Control and Data Acquisition (SCADA) and Programmable Logic Controller (PLC) devices on the shop floor. These devices are programmed with their low-level coding languages. This presents several drawbacks, such as inconsistencies and naming errors between the design and the implementation steps, or difficulties in integrating new Industry 4.0 functionalities. This paper presents a design and management methodology, and a software architecture to overcome these drawbacks. The objective of the methodology is the interconnectivity of domain knowledge, software, and hardware components to automatically generate Industry 4.0 Cyber-Physical Systems (CPS) for industrial processes. The methodology is composed of five main steps: designing the process, programming the PLC, capturing data, managing the process and visualizing it. Based on the methodology and the architecture, a set of tools targeting ISA-88 processes has been implemented and validated. IEC-61512 (also known as ANSI/ISA-88) is a standard addressing batch process control. It follows a design philosophy for describing equipment and procedures, equally applicable to manual processes. The methodology has been validated on a process controlled by a Siemens 1200 PLC. The main advantages of this methodology identified during the validation are: (i) reduction in the time required to design the ISA-88 process, (ii) reduction in the PLC programming time and associated errors, (iii) automatic integration of a CPS with data capture and visualization functionalities, and (iv) remote management of the process.

**Keywords:** CPS; Industry 4.0; OPC UA; ISA-88

## 1. Introduction

The Industry 4.0 paradigm relies on the availability of data about industrial processes to digitalize manufacturing lines and generate newly added value services. Cyber-Physical Systems (CPSs) are key elements of the Industry 4.0 paradigm: they represent the union of the virtual and the physical world. The integration of CPSs into manufacturing lines leads to Cyber-Physical Production Systems (CPPSs)—mechatronic systems (physical part) monitored and controlled by software brains and digital information (cyber part) [1,2].

However, the myriad of Information Technology (IT) technologies, standards and specifications related to Industry 4.0 create a knowledge barrier to integrate them into the design and deployment cycles of production processes: these IT technologies follow a completely different philosophy from the regular tools used by Operations Technology (OT) engineers. This creates a separation between industrial domain knowledge, OT hardware and software knowledge, and Industry 4.0 IT-based software knowledge.

Although bigger companies may afford multidisciplinary teams with the required domain, IT and OT knowledge and experience, this is not the case for most end user or automation engineering from small and medium-sized enterprises (SMEs). Moreover, traditional automation SMEs, which have deep knowledge about the OT world, are facing

more demanding customer requirements to capture and export data from production lines and machines using Industry 4.0 technologies. The lack of IT expertise is a burden that may consume a relevant portion of the project budget and dismisses the competitiveness of SMEs. Low-cost alternatives to implement Industry 4.0 concepts and technologies are required [3].

This paper proposes a general methodology, a software architecture, and a set of tools validated on the ISA-88 batch process domain to overcome this situation, connecting industrial process domain knowledge with IT and OT technologies. Although the development of these tools requires IT knowledge, once they are deployed, they only require automation engineers to receive basic training on a Web application to integrate the methodology into designing and programming OT workflows to automatically integrate Industry 4.0 CPS functionalities.

In traditional automation programming workflows, after some internal design meetings, the automation engineer creates a new Programmable Logic Controller (PLC) program and manually inserts the names and memory addresses of the variables. Then, this information is shared with the rest of the team. IT engineers use this information to connect to the PLC and access the values of the variables. This process is time consuming and error prone. With the proposed methodology, the design of the process is defined on an intuitive Web interface customized to the industrial domain of the automation engineers. Then, automation engineers are automatically provided with an initial version of the PLC program containing the names and memory addresses of the variables. Once the PLC is running, the variables are automatically captured, stored, and visualized, saving time and avoiding naming errors.

The objective of the methodology is twofold: (i) to ease and speed up the design and automation programming steps of industrial projects and (ii) to automatically generate a CPS of the processes with data visualization and process management functionalities integrated, serving as a solid foundation for future Industry 4.0 services such as process simulations or predictive maintenance capabilities.

The methodology is composed of five steps. First, the design step models and defines the relevant elements of the processes. Once the design is completed, the variables and the memory position at the shop floor devices are automatically defined. Second, this information is directly imported into a skeleton of a real program of regular automation tools to ease and fasten the programming and deployment of the physical processes and avoid naming errors. After that, once the industrial process is running, the variables and methods of each element of the design are published following their design name, and a monitoring module captures the value of these variables to store it in a database. Moreover, in this step, a virtual representation of the industrial processes and an OPC UA server to read and write their variables are automatically generated. Thus, a CPS of the industrial processes connecting the virtual and the real processes is available without additional engineering efforts. In the fourth step, the processes are managed using Web tools. On the fifth step, a synoptic is designed using a Web tool, to visualize the industrial production processes.

Although the methodology is applicable to any industrial domain, the IT tools should be customized to meet the requirements of specific industrial domain workflows. Thus, Web tools have been developed and validated with a proof of concept (PoC) of an ISA-88 batch process controlled by a real PLC. ES Solidos Process Engineering, a process automation engineering SME, has been the promotor of the work described in this paper, leading the project and contributing to the technological development with this automation and ISA-88 knowledge to set the requirements, and to validate the results designing the batch process and programming the corresponding PLC.

The reduction in process designing time, PLC programming time, and naming errors has been detected as a positive effect of the validation of the PoC. Moreover, the main advantages that have been identified are the added value of the Industry 4.0 functionalities

to capture data, visualize data and manage the process from any current Web browser with no additional effort.

This paper is organized as follows. Section 2 reviews the state of the art. In Section 3, the design methodology is presented. Section 4 and 5 focus on the ISA-88 use case and its validation. Finally, in Section 6, some conclusions and possibilities for future work are presented.

## 2. Literature Review

### 2.1. CPS

The German government presented the Industrie 4.0 term in 2011. The objective of the fourth industrial revolution is to work with a higher level of operational productivity and efficiency, connecting the physical world to the virtual world. Industry 4.0, also known as the Industrial Internet of Things (IIoT), is related to several technologies, such as Internet of Things (IoT), Industrial Automation, Cybersecurity, Intelligent Robotics, and Augmented Reality [4].

The term cyber-physical systems (CPS) was coined in the USA in 2006 and has received several definitions [5]. CPS is the merger of "cyber" as electric and electronic systems with "physical" things. The "cyber component" allows the "physical component" (such as mechanical systems) to interact with the physical world by creating a virtual copy of it. This virtual copy will include the "physical component" of the CPS (i.e., a cyber-representation) through the digitalization of data and information [4].

In general, a CPS consists of two main functional components: (i) the advanced connectivity that ensures real-time data acquisition from the physical world and information feedback from cyberspace; (ii) intelligent data management, analytics and computational capability that constructs the cyberspace [6]. In [7], the authors review the literature about CPS identifying challenges, approaches and techniques for Industry 4.0 CPS.

Several attempts for converting traditional industrial systems into interoperable, digitalized components have been made during the last years [4]. Nevertheless, the main challenges for both Industry 4.0 and IIoT, including security, standard exchange of data and information between devices, machines and services, remain open research areas [8]. In [9], authors present a survey of edge computing for industrial applications, identifying proposed architectures, advances and challenges.

Existing international standard reference architectures, Reference Architectural Model Industry 4.0 (RAMI 4.0) and Industrial Internet Reference Architecture (IIRA) provide reference models that are not straightforward to implement [10]. Moreover, architectures proposed in the literature follow ambitious approaches covering several functionalities (orchestration, load balancing, advanced security models...) required for a complete Industry 4.0 implementation.

For example, back in 2015, Ref. [6] proposed a general architecture for manufacturing systems based on a five-level CPS (smart connection level, data-to-information conversion level, cyber level, cognition level, and configuration level) to serve as a practical guideline for future implementations. Since then, researchers have proposed different architectures to implement CPS.

In [11], the authors present a service-based architecture for the interaction of control and manufacturing execution systems in Industry 4.0 and IoT-cloud architecture for smart systems. In [12], researchers present a computing platform for CPS based on Docker, deploying it at their laboratory. In [13], authors present an infrastructure for a software-defined cloud manufacturing use case. In [10], the authors present an example of transforming stand-alone equipment into an OPC UA enhanced CPS.

These general architectures require complex implementation, deployment, and customization efforts, out of reach for SMEs focused on industrial process design and implementation. The architecture proposed in this paper focuses on these use cases.

*2.2. ISA-88 Introduction*

ISA-88, also known as S88 or IEC-61512 standard, defines a batch process as: "A process that leads to the production of finite quantities of material by subjecting quantities of input materials to an ordered set of processing activities over a finite period using one or more pieces of equipment".

ISA-88 defines three main models to describe batch processes: process, physical and procedural control. The process model refers to the chemical and physical changes to the materials (product). It divides the batch process into a smaller subdivision of the process: process into process stages, process stages into process operations, and process operations into process actions.

The physical model focuses on the control of the batch process in terms of hardware, dividing a plant into seven levels: enterprise, sites, areas, process cells, units, equipment modules, and control modules.

Finally, the procedural control model focuses on the control of the batch process in terms of software, describing how it should be conducted. It is divided into procedures, unit procedures, operations, and phases. Procedures define the strategy for making the batch. Unit procedures consist of an ordered set of operations; they define which operation will be run by each unit. Operations represent a change (physical or chemical) to the material. Finally, phases are the lowest level in the procedural control model.

ISA-88 also defines states and commands (Figure 1). The states define the condition that procedural and equipment elements can be in at any time. Transitions between states may either happen automatically as part of the manufacturing process or be forced by commands. States may be final or transient (finish with "ing").

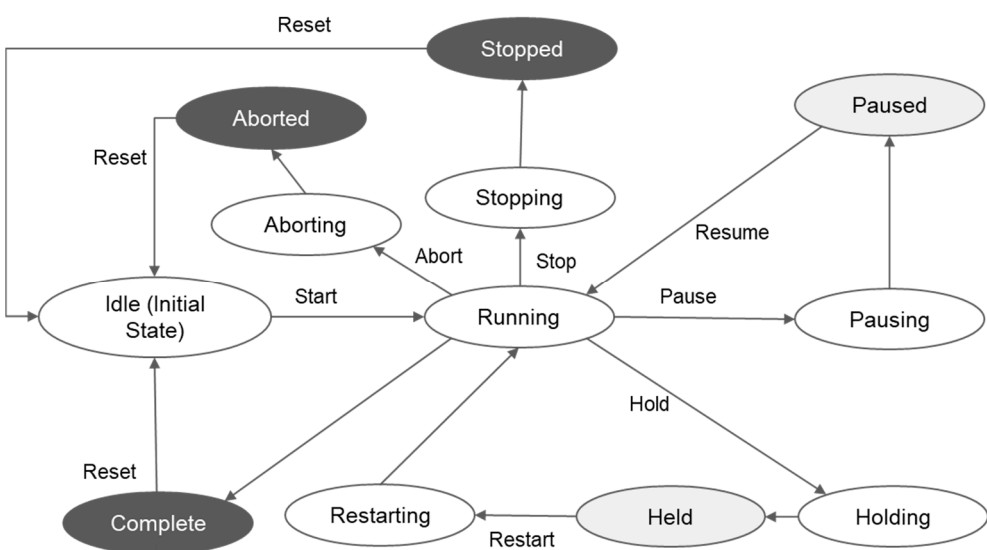

**Figure 1.** States defined by ISA-88 (source: [14]).

The relationship between the three models is illustrated in Figure 2. The procedural control model represents what you want to produce, the process model how you produce it, and the physical model what you use to produce it.

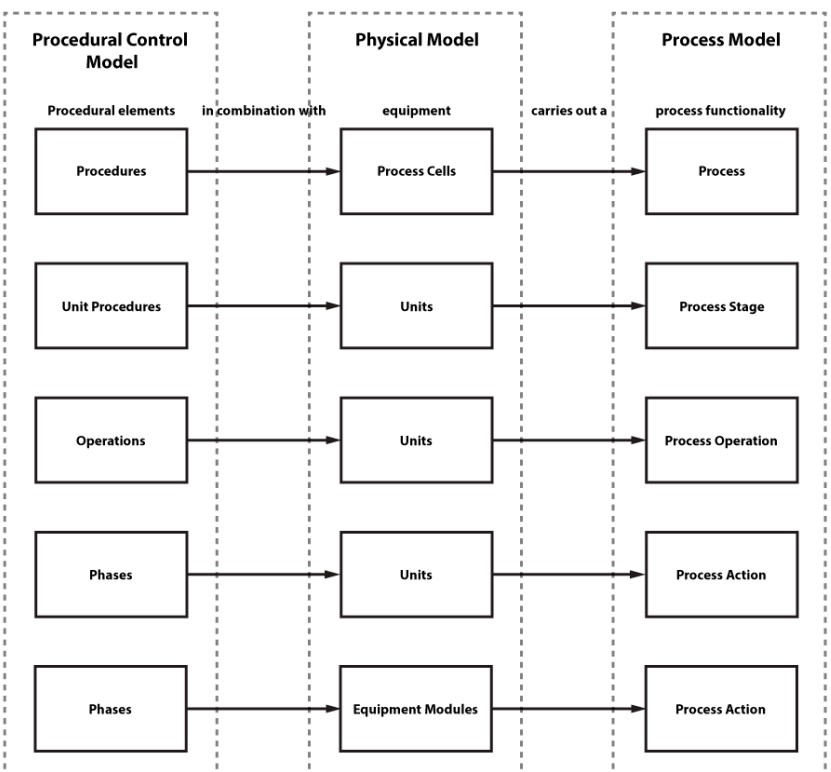

**Figure 2.** ISA-88 layer interconnection (source: https://www.plcacademy.com/isa-88-s88-batch-control-explained/, accessed on 1 April 2022).

### 2.3. Software Tools for ISA-88 Process Design and Process CPS

Vyatkin [15] identifies the growing relevance of software engineering in automation. In [16], the author reviews practices related to software engineering, methods, and tools used in the development life cycle of automation applications. He concludes that software engineering tools and techniques were still at an early stage and identifies the need for analysis and design methodologies, as well as support tools, that abstract the user from the underlying technologies. This paper responds to this need, which is still present.

Alvarez et al. [17] present a model-driven development approach based on the standardized GRAFCET modelling language (IEC 60848) and GEMMA. They propose a framework implementing an incremental development process that allows different engineering profiles to work through analysis, design, and implementation phases. Some of the main advantages they identify are related to reducing design errors, providing easier verification procedures and structured designs, improving productivity, and reducing development time through model reuse and code generation.

Focusing on general design methodologies, more focused on the internal programming of the PLC, Barbieri and Gutierrez [18] present a methodology to couple PLC and digital modes, setting the vocabulary and the management of the operational modes of industrial automation systems. The methodology proposes the standardized GRAFCET modelling language (IEC 60848) to apply a hierarchical design pattern to mimic the PLC software.

The adaptation of these proposals to ISA-88 processes would be a complex process and it could not be seamlessly integrated into current workflows or integrated CPS.

In [19], the authors propose OPC UA-based tools to access field data in automation systems, proposing a graphical interface to assist the creation of OPC UA servers acting as CPS for ISA 95/88 processes. However, these tools do not support the design of the processes, assist in programming the PLC controlling the processes, or the visualization or management of the process.

Kaczmarczyk et al. [20] presented SkuBATCH, a customized batch management system for teaching purposes connected to a PLC controlling a particular process and using

OPC UA for communications. The system process control has been designed and implemented based on ISA-88 and ISA-95 standards. They identify requirements and workflows of the system but do not provide design functionalities, methodologies, or implementations.

Finally, previous approaches lack acceptance in the industrial practice of PLC programming since they are based on modelling languages that are usually not familiar to the designated users [21]. New methodologies, architectures and implementations are required to ease and automate the integration of CPS into manufacturing lines in general. This paper proposes a methodology and software tools validated on a special use case, ISA-88 processes, to assist the design and implementation of industrial processes and automatically generate a CPS integrating Industry 4.0 functionalities.

### 3. Design Methodology

The objective of the proposed methodology is twofold: (i) to ease and speed up the design and the later programming of the automation of the industrial process and (ii) to automatically generate a CPS with the information required to capture and store data from the process and manage and visualize it.

The methodology is composed of five steps (Figure 3): design, program, capture, manage and visualize. The first step focuses on the design of the process. Within this step, the different elements of the process must be defined. Moreover, information about the real PLCs (IP address, port, variable list...) that will control the process is also required to later merge the virtual design of the process with the real running process. At the program step, the design information is used to automatically generate a skeleton of the program of the PLC that will control the real process. This skeleton serves as a template and contains the definition of the data blocks of the PLC program where variables will be stored. The automation engineer is responsible for programming the PLC based on the created skeleton.

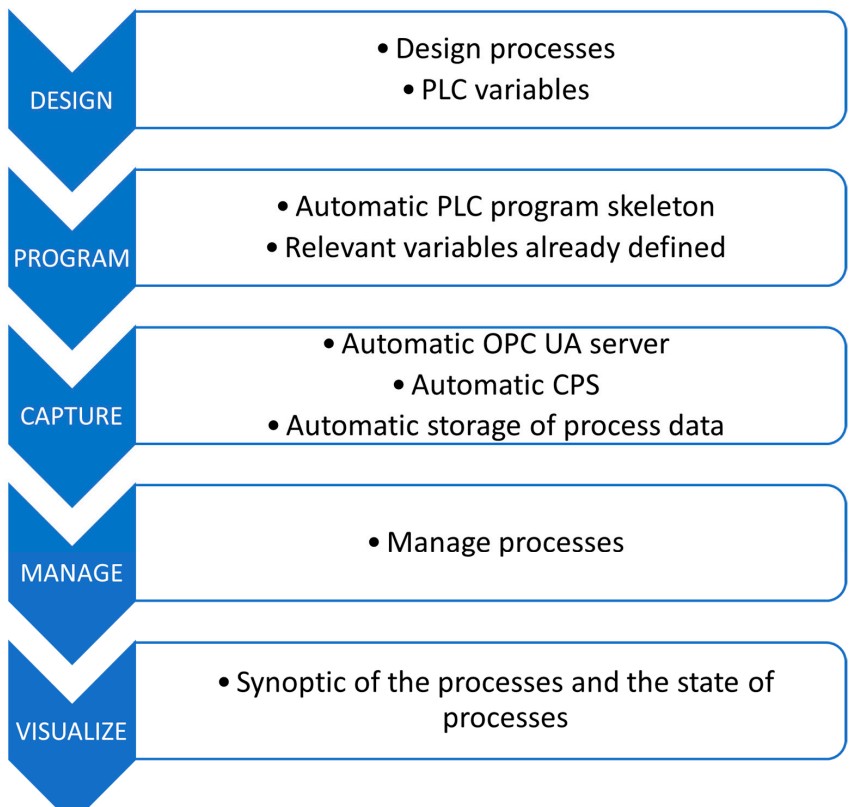

**Figure 3.** Summary of the methodology (source: authors).

Once the PLC is running and deployed, within the capture step a CPS of the process is generated automatically. This CPS integrates an OPC UA Server accessing the PLC and

publishing the values of the variables defined at the design step. These variables will also allow for writing their values from the OPC UA clients. Moreover, an OPC UA client is also generated automatically to subscribe to process variables and store them in a database. Finally, at the managing step, a Web application provides the interface to start and stop recipes and procedures, and at the visualization step, a synoptic of the process is presented.

Figure 4 presents a general software architecture to implement the methodology. Numbers next to each component represent the steps of the methodology where the components are involved. The Web designer, process manager and synoptic follow a traditional Web application architecture with a database to persist data and a server to provide HTTP REST API endpoints.

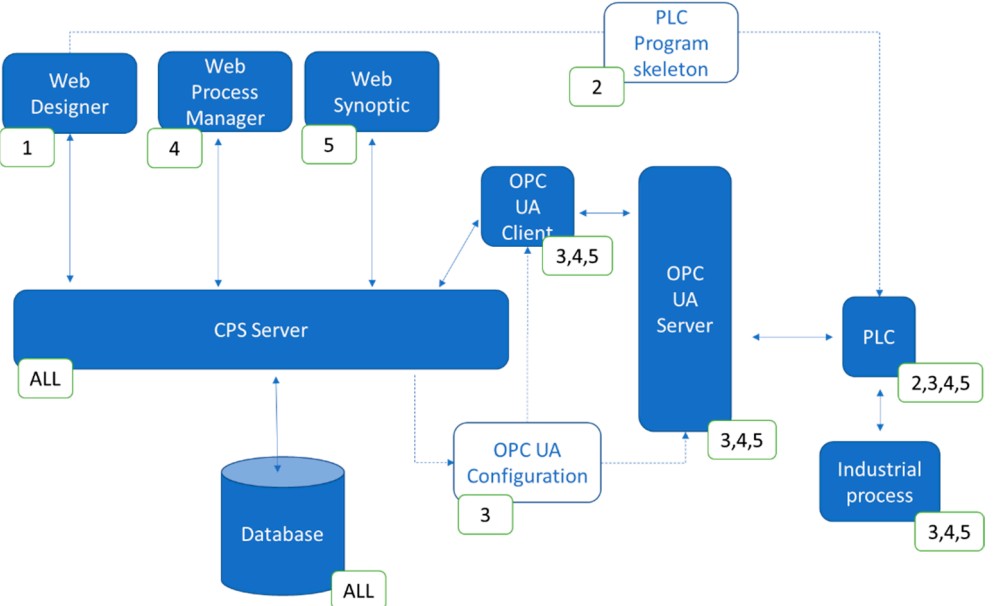

**Figure 4.** General view of the architecture (source: authors).

Once the design of the process is over, the server creates (i) a skeleton of the program of the PLC containing the definition of the relevant variables of the industrial process, and (ii) the configuration data required to automatically generate the OPC UA server and client. After the PLC is programmed and controls the industrial process, the OPC UA server connects to the PLC to read and write the value of the variables. The OPC UA client connects to the OPC UA server, sends values of the variables to the server to be persisted, and receives new values of control variables to be sent back to the PLC through the OPC UA server. The Web manager and the synoptic interact with the server, which reads data from the database and when it is required updates variable values through the OPC UA client.

## 4. ISA-88 Use Case

The methodology and the software architecture could be applied to any industrial domain. However, the presented software tools and the validation have been customized to meet ISA-88 batch process requirements (Figure 5). Although this customization for a specific domain requires IT knowledge, once the tools have been developed OT engineers can integrate them into their workflows and use them without IT expertise. These tools could be customized with requirements from other industrial processes and domains to automatically generate CPS and efficiently assist automation engineers on the design and management of the processes.

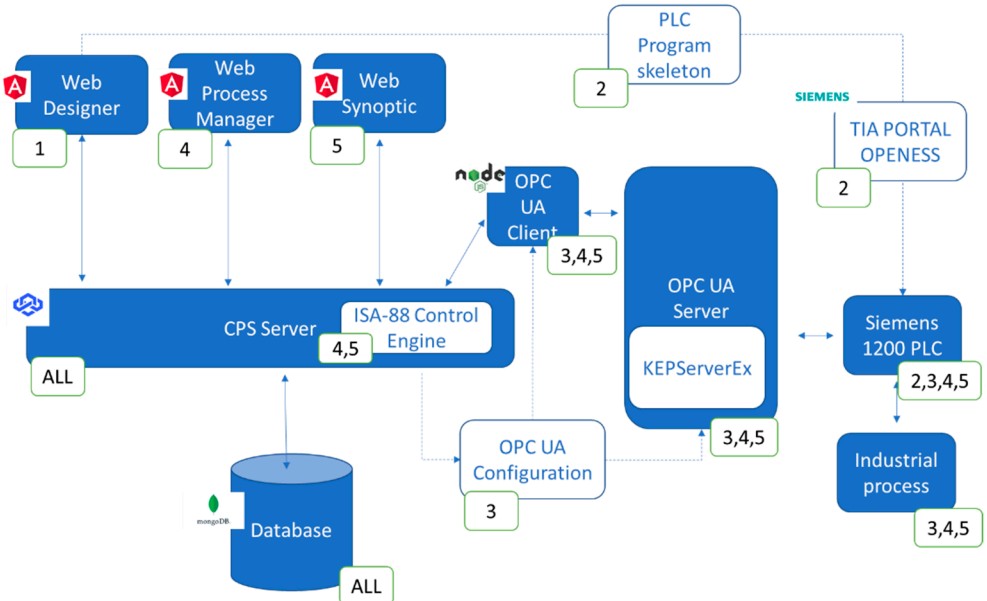

**Figure 5.** Architecture for the ISA-88 use case (source: authors).

ISA-88 defines any discrete manufacturing process by following a step-by-step procedure that can be understood by people who work with this kind of process. The full description of ISA-88 is not part of the scope of this paper, but an extended description of it can be found in references from previous sections.

Regarding the core of the technology stack, the Web applications have been implemented using Angular, a framework from Google, and the UI component library PrimeNG. The CPS server has been implemented in Node.js, based on the Loopback framework from IBM. The database is based on the NoSQL MongoDB database. The following subsections describe the details and the customization of each step for the ISA-88 use case.

### 4.1. Design Step

Within the first step of the methodology, four types of ISA-88 elements are defined to establish a batch manufacturing process unequivocally following a ladder paradigm. This paradigm is also applied to commercial batch design tools such as Proficy Batch from Fanuc Automation, Factory Talk Batch from Rockwell Software, or Inbatch from Wonderware. Engineers can create blocks from these elements to be later reused in different projects. These elements are as follow:

- Physical models define the real mechanical elements that take part in the process (tanks, valves, temperature controllers...). These elements are called control modules and have some equipment and phase variables associated that define their state, for example, a valve can be defined by a Boolean variable that is True if it is open or False otherwise. The control modules are classified using a tree structure: a control module is part of a bigger machine that performs a function (equipment module). At the same time, several machines work together to make a partial product of the process (unit). Finally, units combined create the final product (cell).
- Area models define the actions (phases) conducted by the equipment modules. First, phase variables that represent the action that is going to take place in that phase are described. For example, in a mixer, the actions of adding liquids by "the current amount of liquid" and "the amount of liquid objective" in the mixer are represented. The value of these variables is not yet assigned, it is only stated that they are needed (Figure 6).
- Procedures, once the physical and area models are defined, must be arranged to create a certain final product by setting their execution order. The arrangement and order information are described inside a procedure.

- Recipes represent the final step of the process where all the phases inside an area model are arranged and the variables are given certain values to obtain a specific output from the final product. For example, given some equipment and some actions, by changing the amount of "Element A" added or the amount of time an action is repeated, a different outcome could be achieved in batch production (Figure 7).

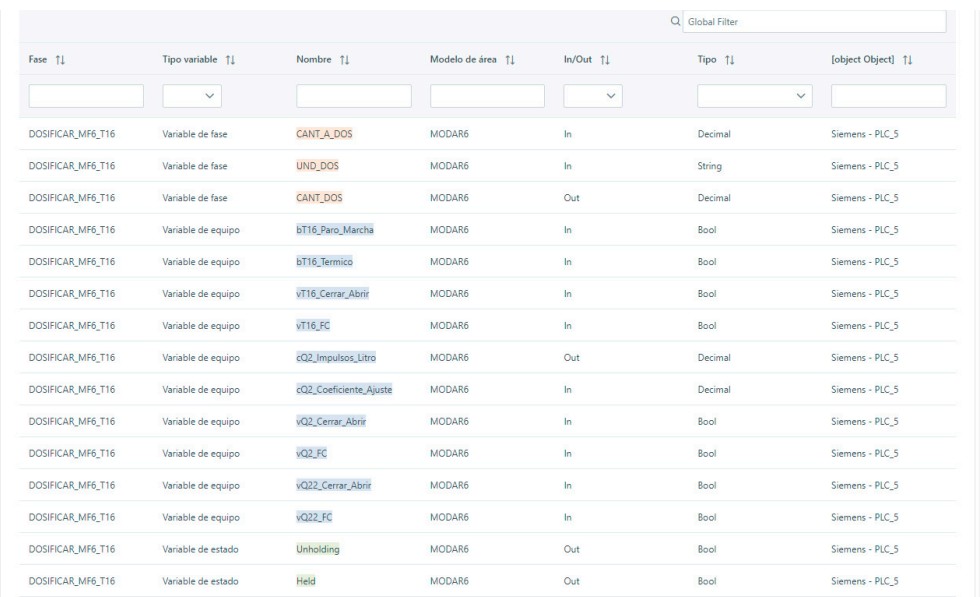

**Figure 6.** Example of a list of variables of a process (source: authors).

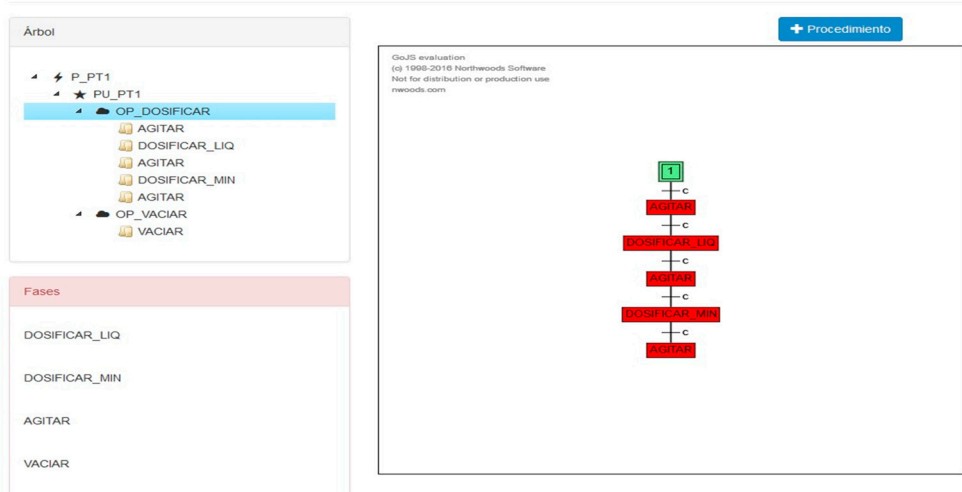

**Figure 7.** Recipe example (source: authors).

### 4.2. Program Step

Based on the design information, the server creates a skeleton of the program for the PLC. The skeleton consists of a list of variables that include the equipment variables, the phase variables, the states, and the commands for each phase (each phase must have its states and commands).

Each PLC manufacturer has its methodology to create and transfer this skeleton to a PLC. For Siemens PLCs, this is based on the Openness API. Openness defines the format of XML files that can be imported into new TIA Portal projects to initialize the project with certain elements (configurations, functions, data blocks...). Thus, the server creates an XML file with the definition and the memory position of relevant variables and then this file

is imported into a new TIA Portal project. Variables are automatically assigned memory positions of the PLC, starting from the data block address specified by the engineer. Then, the engineer programs the PLC to control the batch process using these variables. The main sections of this XML file are:

- A description of the versions of TIA Portal and Openness that are going to be used to import this XML file.
- One section ("SW.Datablock") per phase described on the area model. The main subsections for each phase are as follows:
    - o   One subsection contains the name and type of the variables of that phase. Since the memory position is not a parameter, TIA Portal assigns the value on its own. The memory position is calculated considering how TIA Portal makes this assignment according to the memory required for each variable type and their order in this XML document;
    - o   The name of the phase;
    - o   The number of the data block.

An example of a summary of a section of the XML loaded into TIA portal is shown in Figure 8.

```xml
<SW.DataBlock ID='2'>
    <AttributeList>
        <AutoNumber>false</AutoNumber>
        <DatablockType>SharedDB</DatablockType>
        <Interface>
            <Sections xmlns='http://www.siemens.com/automation/Openness/SW/Interface/v1'>
                <Section Name='Static'>
                    <Member Name='CANT_A_DOS' Datatype='Decimal'/>
                    <Member Name='CANT_DOS' Datatype='Decimal'/>
                    <Member Name='UND_DOS' Datatype='String'/>
                    <Member Name='bT17_Paro_Marcha' Datatype='Bool'/>
                    <Member Name='Suspend' Datatype='Bool'/>
                    <Member Name='Unhold' Datatype='Bool'/>
                    <Member Name='Unsuspend' Datatype='Bool'/>
                </Section>
            </Sections>
        </Interface>
        <MemoryLayout>Standard </MemoryLayout>
        <Name>DOSIFICAR_MF6_T17</Name>
        <Number>14</Number>
        <ProgrammingLanguage> DB </ProgrammingLanguage>
        <Type> DB </Type>
    </AttributeList>
</SW.DataBlock>
```

**Figure 8.** Section of XML file loaded into TIA Portal (source: authors).

*4.3. Capture Step*

At this point, since all the information of the process is already defined, the methodology applies it to generate a CPS to control the process, capture the information and provide the visualization. In order to communicate with the PLC, KEPServerEX has been installed. KEPServer is a widely installed industrial connectivity platform that provides a single source of industrial automation data. It has a REST API to create OPC UA servers connecting to PLCs automatically. Thus, based on the ISA-88 process design data (the variables and their memory position in the PLC), the CPS server automatically configures the OPC UA server with all the variables needed to control the process. Moreover, KEPServer has a myriad of drivers to connect to PLCs from different providers (Siemens, Beckhoff, Allen-Bradley...) maintaining the same API. Thus, integrating a PLC from a new provider would require no code or minor changes.

Finally, the CPS server also automatically generates an OPC UA client based on the node-OPC-UA library to communicate with the server to read and write variable values.



The information extracted by this client is being stored in a MongoDB database so it can be analyzed or visualized later.

*4.4. Manage Step*

The ISA-88 control engine is based on the integration of the information on the recipes of the design phase. This allows the whole ISA-88 process to be controlled, since the order where the phases are executed and the variables that define that process are already stored in the database. Controlling an ISA-88 manufacturing process is a complex task. In a real scenario, the control of the ISA-88 process should be performed by a certified Supervisory Control and Data Acquisition (SCADA) system. The implemented control functionalities have not been certified to fulfil the real-time and security requirements of real scenarios; they provide a functional implementation to validate the methodology. However, even without the control system, the methodology provides relevant value: data capture and visualization functionalities, reduction in design time, PLC programming time and human errors.

Once the process is running, the Web manager presents a schematic view of the different procedures and recipes to manage the process and as a representation of the status of the process (Figure 9).

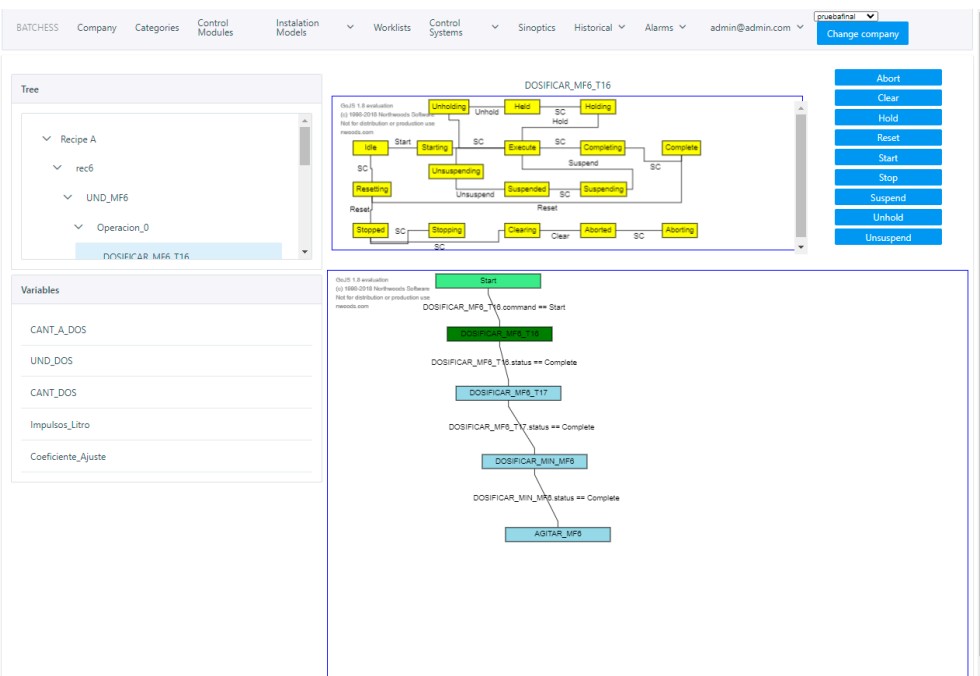

**Figure 9.** Execution diagram of a recipe (source: authors).

*4.5. Visualization Step*

In real-life situations, views of the processes are usually represented on the shop floor using dynamic representations of the process, known as synoptics. Since each process is different, synoptics must be custom-made to provide the best representation for each scenario. To do that, the Web synoptic editor allows for drawing a symbolic representation of the process using both components from the SymbolFactory library, custom shapes, texts, and images. Values, colours, and fonts of the elements of the synoptics can be linked to react to the value of the variables of the processes. Once a synoptic has been defined, it visualizes the status of the process as a regular SCADA view of regular software such as WinCC from Siemens or SCADA Crew from ESA (Figure 10).

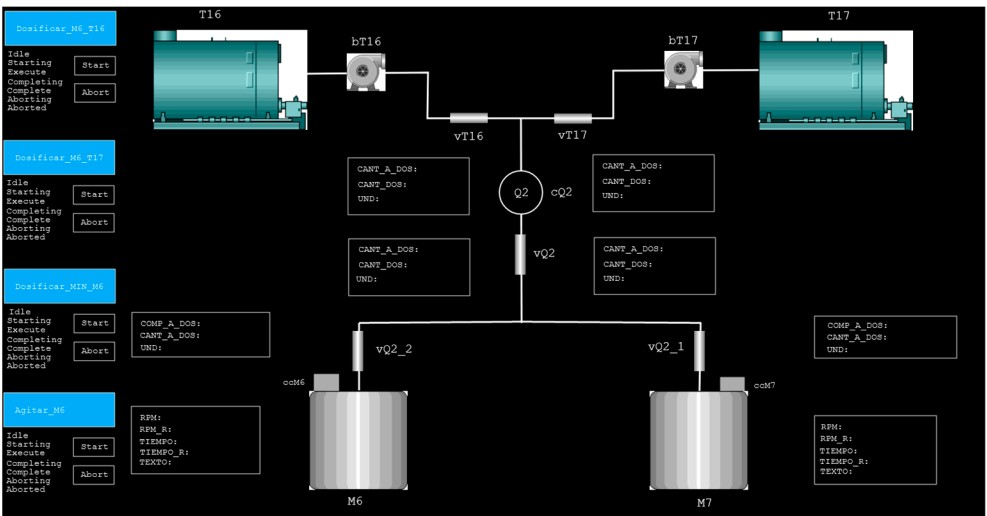

**Figure 10.** Synoptic of the validation process (source: authors).

## 5. Validation

The validation has been based on a PoC of a simplified ISA-88 process resembling a real process (Figure 11) designed by two automation engineers of ES Solidos. This process is composed of one cell with a single unit and four equipment modules: two dispensers (each one with three valves, a pump and a container as control modules), a mixer (with a single control module) and a single loading gate (also as a single control module each). Loading gate M2 was not used for the validation.

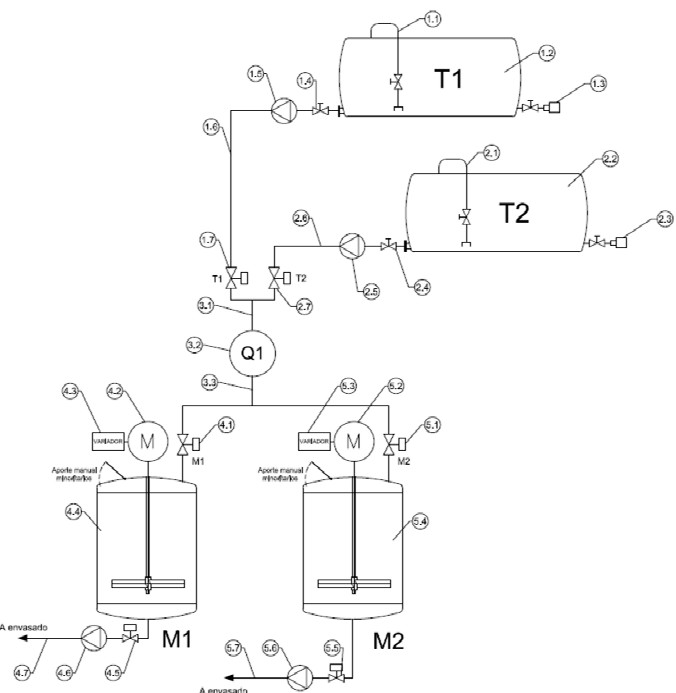

**Figure 11.** Footprint of the validation batch process (source: authors).

Once the physical model and the area model have been designed using the Web application by automation engineers, the skeleton of the program for the PLC has been generated. This skeleton has been imported into a TIA Portal project to program a Siemens 1200 PLC. Instead of connecting to real components, the PLC has been programmed to simulate the input and output signals of real processes.

To control this process, two recipes have been designed. The main differences between them are the amount of material disposed of by the dispenser T1 and T2, the time the mixer is turning and its speed, and the order in which the dispensers are opened (first T16 and then T17 or the other way around). Synoptic dashboards of the validation process have been generated with the provided Web synoptic tool (Figure 10).

The provided tools have been validated by two process engineers from ES Solidos. The claims of the results of the validation are based on the expertise and knowledge of these engineers, who have not been directly involved in the related research project. These engineers have successfully designed the process, programmed the PLC, designed the synoptic and validated the results.

Even though the same process has not been programmed with the traditional workflow, where the design of the process is done on paper and then the programming is done directly in TIA Portal, the feedback from the validation has been positive. There have been 0 naming errors. Engineers do not know how many naming errors would have been introduced while programming the TIA Portal program directly, but to identify and fix each of them would have cost several hours. program with the variables already named and their memory position assigned has saved Moreover, having a skeleton of the TIA Portal around one hour of time.

Although the methodology must be further validated to confirm the claims of the automation engineers involved in the validation of the PoC, the obtained positive result encourages further validation efforts.

## 6. Conclusions

This work proposes a general five-step methodology, a software architecture, and a set of tools to interconnect industrial process domain knowledge with IT and OT technologies and automatically generate Industry 4.0 CPS.

The methodology is composed of five steps. First, the design step models and defines the relevant elements of the processes. Once the design is complete, the variables and the memory position at the shop floor devices can automatically be defined. Second, this information is directly imported into a skeleton of a real program of regular automation tools to ease and speed up the programming and deployment of the physical processes and avoid naming errors. Once the industrial process is running, the variables and methods of each element of the design are published following their design name, and a monitoring module captures the value of these variables to store it in a database. Moreover, within this step, a virtual representation of the industrial processes and an OPC UA server to read and write their variables are automatically generated. Thus, a CPS of the industrial processes connecting the virtual and the real processes is available without additional engineering efforts. In the fourth step, the processes are managed using Web tools. On the fifth step, a synoptic is designed using a Web tool, to visualize the industrial production processes.

Although the methodology is applicable to any industrial domain, the IT tools should be customized to meet the requirements of specific domain workflows. Thus, Web tools have been developed and customized for a validation of a PoC of an ISA-88 batch process. This set of software tools, following the proposed methodology and architecture, have been provided to automation engineers without IT knowledge for the validation. The validation has been based on the design and programming of a simplified ISA-88 process resembling a real process and programmed into a Siemens 1200 PLC simulating input and output signals. Two automation engineers from ES Solidos have successfully designed the process, programmed the PLC, designed the synoptic and validated the results.

The reduction in process designing time, PLC programming time, and naming errors has been detected as a positive effect of the methodology during the validation. These positive results encourage further validations to confirm the claims of the two automation engineers that have taken part. Moreover, the main advantages that they have identified are the added value of the Industry 4.0 functionalities to capture data, visualize data and manage the process from any current Web browser with no additional effort. These

functionalities are out of the expertise of the automation engineers, and currently their integration on automation projects by ES Solidos and its clients (manufacturing final users) requires external technology providers.

The main limitation of the methodology is the customization required to adapt the IT tools to the requirement of each industrial domain. Engineers with OT knowledge must have tools adapted to their domains and their workflows to seamlessly integrate the methodology and the automatic CPS into their automation processes. This customization is required once per industrial domain and requires OT SMEs to have IT providers. Once this customization is performed, it is also valid for other SMEs of the same domain. Moreover, IT tools should be updated periodically to adapt to technological updates of other tools of the development ecosystem, such as new versions of TIA Portal, or to include security patches of the software components.

The main future line of work is focused on validating the methodology and the architecture customizing the software tools to integrate knowledge from different industrial domains. It is very difficult to get the approval of a final user manufacturing real products to interfere with the real production process. Thus, the compromise solution has been the programming of a PoC for the validation. The PoC consists of a PLC simulator, where the outputs are controlled through the PLC program but not connected to the real word. Thus, we are actively looking for funding opportunities involving final manufacturing users to extend the validation to real manufacturing scenarios, new domains, and more automation engineers.

Finally, updating the software components managing the industrial process to reduce the gap with functionalities provided by SCADA systems (real-time requirements, resilience...) has been identified as an added value feature by ES Solidos engineers.

**Author Contributions:** Conceptualization, A.G., X.O. and I.V.; methodology, A.G., X.O., I.V.; software, X.O., A.G., U.A.; validation, X.O., U.A. and I.V; resources, I.V.; writing—original draft preparation, A.G. and X.O.; writing—review and editing, U.A.; visualization, X.O. and U.A.; supervision, A.G.; project administration, X.O.; funding acquisition, X.O. and A.G. All authors have read and agreed to the published version of the manuscript.

**Funding:** This research was partially funded by the Hazitek program of the department of industry of the Basque Government (IG-2015/00299 and ZL-2018/00518).

**Institutional Review Board Statement:** Not applicable.

**Informed Consent Statement:** Not applicable.

**Data Availability Statement:** Not applicable.

**Conflicts of Interest:** The authors declare no conflict of interest.

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
