# Peer review of "Methodology and Tools to Integrate Industry 4.0 CPS into Process Design and Management: ISA-88 Use Case"

_information, doi:10.3390/info13050226_

Round 1

Reviewer 1 Report

This submission looks more like a senior design project description that it does a research paper. The paper purports to address the issue that SMEs have OT expertise but not IT expertise. However, the solution described here requires a substantial amount of IT expertise. So the solution described here would not address this issue.

In addition, the company that is involved with the project, ES Solidos Process Engineering, is a solutions provider. They are not an end user SME.

The literature section is very weak. There doesn't seem to be a very good grasp of industry 4.0. There really isn't any tie-in with their solution and industry 4.0. Inclusion of such things as fog computing has nothing to do with the type of project that they're presenting. The authors have no familiarity with Digital Twin technology, which provides the framework of the virtual development to physical realization that they seem to be trying for.

The paper does describe the project implementation. However, it seems very ad hoc as opposed to something that can be generalized. Additionally, the IT expertise needed here is still pretty unreachable by SMEs as the ability to close this gap for SMEs was defined as a major reason for this effort.

The validation doesn't really seem to be a validation. As the paper states, "instead of connecting to real components, the PLC has been programmed to simulate the input and output signals overall processes." The claim of validity from process engineers at the ES Solidos Process Engineering is provided as a statement without any real backup. The fact that a company individual is an author of the paper makes the claim even more suspect. It clearly isn’t validation done by a disinterested third party.

In the conclusion, there is a claim that "reduction in process designing time, PLC programming time, in naming errors have been detected as positive facts a methodology during validation." However, there is neither any quantification of this nor any actual support of this claim.

Finally, it is very unclear as to what the take away of this paper should be, other than they did some programming and think it was beneficial. The paper needs a lot more work.

Author Response

This submission looks more like a senior design project description that it does a research paper.

The paper proposes a novel methodology, software architecture, and a set of tools targeting ISA-88 processes to update industrial automation process design and programming to automatically include Industry 4.0 functionalities. Although the implementation of the software tools has required senior project design, management and programming abilities, the main contribution of the paper is the methodology and the architecture, not the specific software tools targeting the ISA-88 validation.

The paper purports to address the issue that SMEs have OT expertise but not IT expertise. However, the solution described here requires a substantial amount of IT expertise. So the solution described here would not address this issue.

We have updated the explanation of the purpose to make it clearer that the objective is to provide SMEs with software tools that once developed, allow SMEs to use them only with OT knowledge. Following the methodology, these tools automatically capture industrial process data and generate CPS of the process without IT expertise.

In addition, the company that is involved with the project, ES Solidos Process Engineering, is a solutions provider. They are not an end user SME.

Although they are solution providers, Es Solidos is an automation engineering SME with very limited IT knowledge. They are focused on the design, programming and deployment of OT automation projects of batch processes of end users SMEs. We have updated the text to make this point clear.

The literature section is very weak. There doesn't seem to be a very good grasp of industry 4.0. There really isn't any tie-in with their solution and industry 4.0. Inclusion of such things as fog computing has nothing to do with the type of project that they're presenting. The authors have no familiarity with Digital Twin technology, which provides the framework of the virtual development to physical realization that they seem to be trying for.

The literature section focuses on CPSs and their integration into industrial processes, specifically  ISA-88 processes. We have removed fog references, as they are not related to the core of the paper, and interested readers have the references to query more details about it. Although we are familiarized with digital twins, we consider that their inclusion will add confusion to the paper. The CPSs generated by the methodology could be considered as a first building block for a digital twin, or even a basic digital shadow. However, as the digital twin term is being used with a very broad meaning lately, we have preferred not to include it on the article.

The paper does describe the project implementation. However, it seems very ad hoc as opposed to something that can be generalized. Additionally, the IT expertise needed here is still pretty unreachable by SMEs as the ability to close this gap for SMEs was defined as a major reason for this effort.

The customization of the software tools has to be done for each domain. The software tools developed for the validation are targeting ISA-88 processes to allow OT engineers to design the process with the software tools easily. Engineers of other domains should be provided with customized tools for their domain. The customization of these tools should be performed by the IT department or their IT provider. SME make use of these tools to integrate CPS without altering their current workflows. We have updated several explanations within the paper to try to make this point clear.

The validation doesn't really seem to be a validation. As the paper states, "instead of connecting to real components, the PLC has been programmed to simulate the input and output signals overall processes." The claim of validity from process engineers at the ES Solidos Process Engineering is provided as a statement without any real backup. The fact that a company individual is an author of the paper makes the claim even more suspect. It clearly isn’t validation done by a disinterested third party.

Although we have tried to fulfil a validation in a real scenario, it is very difficult to get the approval of a final user manufacturing real products to interfere with the real production process. Thus, the compromise solution has been the development of the PLC simulator, where the outputs are controlled through the PLC program but not connected to the real word. Nevertheless, we are actively looking for funding opportunities involving final manufacturing users to extend the validation to a completely real scenario. The same applies to external validations, we are also seeking to find fundings to increase the TRL of the proposal and to extend it to new domains. The claims of the results of the validation are based on the expertise and knowledge of engineers of ES Solidos who have not been directly involved on the related research project. We have extended the future work to include these points.

In the conclusion, there is a claim that "reduction in process designing time, PLC programming time, in naming errors have been detected as positive facts a methodology during validation." However, there is neither any quantification of this nor any actual support of this claim.

Related to the previous point, we are trying to find fundings to extend the validation with more scenarios and users and to correctly quantify the improvements with a proper statistical number of users. Although we have not quantified the improvements on the validation there have been 0 naming errors. We do not know how many naming errors have would the automation engineer introduce while programming the PLC, but to identify and fix each of them would have cost several hours. Moreover, having a skeleton of the TIA Portal program with the variables already named and their memory position assigned, saves around one hour of time. Engineers have only estimated  this time, they have not done it twice (there are around one hundred variables involved on the PLC program of the validation). We have extended the explanation of this point.

Finally, it is very unclear as to what the take away of this paper should be, other than they did some programming and think it was beneficial. The paper needs a lot more work.

We have updated the paper to improve the visibility of the value of the proposal: updating design methodologies and providing OT automation engineers with proper IT tools, it is viable to automatically integrate CPS functionalities into industrial processes, even for SMEs.

Reviewer 2 Report

The papers describes an approach to integrate Industry 4.0 CPS into process design and management. This is a crucial issue in these last years.

Introduction and section 2 introduce and explain in a good manner the analsyed issue. However, the main parts of the paper: section 3 and 4 generate difficulties in understanding what the authors did.

I agree with authors stating "The full description of ISA-88 is not part of the scope of this paper..." and for this reason my suggestions refer to a redesign of section 3 and 4. It could be useful to divide  in paragraphs the proposed argumentation so that it is possible to connect each part of section 4 to the five-step methodology.

For example, you asserted in line 285: "Based on this design information, the server creates a skeleton of the program for the PLC consisting of a list of variables that include the equipment variables, the phase variables, the states and the commands for each phase (each phase must have its states and commands)". A brief explaination about how the server do this coul be useful for the readers in understanding what you have done.

You have use a set of software tools following the proposed architecture and customized to meet ISA-88 batch process requirements but it would be interesting the description abut you did this connection, for example, helped by a flowchart and similar.

Conclusion section is poor in content. What are the approach limitations, advantages and disadvantages.

Please, pay attention in writing the paper: the revision mode of the writing software does not facilitate reading of the paper

Author Response

The papers describes an approach to integrate Industry 4.0 CPS into process design and management. This is a crucial issue in these last years.

We thank and share the perception about the relevance of the topic

Introduction and section 2 introduce and explain in a good manner the analsyed issue.

We have made only minor updates following feedback from another other reviewer,

However, the main parts of the paper: section 3 and 4 generate difficulties in understanding what the authors did.

I agree with authors stating "The full description of ISA-88 is not part of the scope of this paper..." and for this reason my suggestions refer to a redesign of section 3 and 4. It could be useful to divide  in paragraphs the proposed argumentation so that it is possible to connect each part of section 4 to the five-step methodology.

We have followed this proposal to ease the identification of the relation with the steps of the methodology

For example, you asserted in line 285: "Based on this design information, the server creates a skeleton of the program for the PLC consisting of a list of variables that include the equipment variables, the phase variables, the states and the commands for each phase (each phase must have its states and commands)". A brief explaination about how the server do this coul be useful for the readers in understanding what you have done.

You have use a set of software tools following the proposed architecture and customized to meet ISA-88 batch process requirements but it would be interesting the description abut you did this connection, for example, helped by a flowchart and similar.

We have updated the content to include more details about the implementation details to improve the explanation. Now figures include the steps of the methodology where each component is relevant.

Conclusion section is poor in content. What are the approach limitations, advantages and disadvantages.

We have made major changes to this section.

Please, pay attention in writing the paper: the revision mode of the writing software does not facilitate reading of the paper

Please apologize for this error, we have fixed it

Reviewer 3 Report

The article is elaborated on the current topic of integration of chip devices into the CPPS concept. The introduction expresses well the content of the whole article. The current resources are used in an adequate way in the "Literature review" section. Part of the proposed methodology is clearly and concisely presented. The research proposal and its solution are processed very well.

  •  

In the "Conclusions" section, I strong recommend adding a discussion to the proposed solution. The whole level of the article would be significantly benefited by verification on a larger sample of cases or stastically evaluated benefits of the proposed methodology.

Technical comments:

  • The source in Figure 1 is error-labeled as 4. This is its first use and should have a higher number according to the rules.
  • Other images do not list sources. If the images are original, the source "authors" should be mentioned
  • The "Conclusions" section needs to be reformatted.

Author Response

The article is elaborated on the current topic of integration of chip devices into the CPPS concept. The introduction expresses well the content of the whole article. The current resources are used in an adequate way in the "Literature review" section. Part of the proposed methodology is clearly and concisely presented. The research proposal and its solution are processed very well.

We really appreciate the positive feedback from the reviewer 

In the "Conclusions" section, I strong recommend adding a discussion to the proposed solution.

We have included major changes on this section

The whole level of the article would be significantly benefited by verification on a larger sample of cases or stastically evaluated benefits of the proposed methodology.

Although we have tried to fulfil a validation in a real scenario, it is very difficult to get the approval of a final user manufacturing real products to interfere with the real production process. Thus, the compromise solution has been the development of the PLC simulator, where the outputs are controlled through the PLC program but not connected to the real word. Nevertheless, we are actively looking for funding opportunities involving final manufacturing users to extend the validation to a real scenario and more users to identify the benefits of the methodology.

Technical comments:

  • The source in Figure 1 is error-labeled as 4. This is its first use and should have a higher number according to the rules.

Fixed

  • Other images do not list sources. If the images are original, the source "authors" should be mentioned

Fixed

  • The "Conclusions" section needs to be reformatted.

Major changes have been included

Round 2

Reviewer 1 Report

While the authors have improved the paper, there still is a basic problem on what they're claiming. They're making two claims that are still not supported.

The first claim that they have just added is: 51-54 "Although the development of these tools requires IT knowledge, once they are deployed, they seamlessly merge into design and programming OT workflows to automatically integrate Industry 4.0 CPS functionalities."

 however, until they test this out utilizing only OT personnel, this is simply an unsupported claim. That they wish is true.

The second issue is the validation. They simply haven't met the bar for validation.  The authors admit that they looking for funding to do so. Unfortunately, that is an admittance that the validation is incomplete. The fact that they had no naming errors does not support their claims of design validation.

The authors will lead to substantially reduce their claims or provide evidence for them.

Author Response

While the authors have improved the paper, there still is a basic problem on what they're claiming. They're making two claims that are still not supported.

The first claim that they have just added is: 51-54 "Although the development of these tools requires IT knowledge, once they are deployed, they seamlessly merge into design and programming OT workflows to automatically integrate Industry 4.0 CPS functionalities."

 however, until they test this out utilizing only OT personnel, this is simply an unsupported claim. That they wish is true.

We have updated the explanation of the traditional workflow and the proposed methodology workflow to describe their differences. As we have explained on the paper, the IT Web application has been used by OT automation engineers from ES Solidos not involved on the development of the IT tools with no problems. Automation engineers access a regular Web app customized to ISA-88 domain. The automation engineers who participated on the validation had no problem using the app. NNevertheless, related to the next point and the future work, it is completely true we wish we could validate this with a larger automation engineer user base.

The second issue is the validation. They simply haven't met the bar for validation.  The authors admit that they looking for funding to do so. Unfortunately, that is an admittance that the validation is incomplete. The fact that they had no naming errors does not support their claims of design validation.

The authors will lead to substantially reduce their claims or provide evidence for them

We tried to clearly describe the reach of the validation on the previously submitted version. Currently we have no means to extend the validation. Thus, on the latest version we have introduced it as a Proof of Concept (PoC) validation and we have clearly identified that the methodology must be further validated to confirm and generalize the claims of the automation engineers involved on the validation of the PoC at the end of section 5 and within section 6.

Reviewer 2 Report

In the current form, authors have improved the paper quality

Author Response

No response required. Paper has been updated according to the feedback from other reviewers.

Reviewer 3 Report

Please correct epecailly spelling.

Author Response

We have reviewed the paper, including a collegue on the revision. We have fixed errors, updated some sentences, and map styles to the Information Word template.

We hope there are no more mistakes on this version.

Round 3

Reviewer 1 Report

The authors have adequately addressed by concerns by casting this as a proof of concept. It obviously would be useful to have OT engineers use this tool without the involvement of the IT developers. But the paper is adequate for proof of concept.